# Analysis of Drugs in Saliva of US Military Veterans Treated for Substance Use Disorders Using Supported Liquid Extraction and Surface-Enhanced Raman Spectral Analysis

**DOI:** 10.3390/molecules28052010

**Published:** 2023-02-21

**Authors:** Stuart Farquharson, Chetan Shende, Jenelle Newcomb, Ismene L. Petrakis, Albert J. Arias

**Affiliations:** 1Real-Time Analyzers, Inc., Middletown, CT 06457, USA; 2VA Connecticut Healthcare System, West Haven, CT 06516, USA; 3Department of Psychiatry, School of Medicine, Yale University, New Haven, CT 06510, USA; 4Department of Psychiatry, School of Medicine, Virginia Commonwealth University, Richmond, VA 23298, USA

**Keywords:** buprenorphine, patient compliance, supported liquid extraction, surface-enhanced Raman spectroscopy

## Abstract

According to the Center for Disease Control, there were more than 107,000 US drug overdose deaths in 2021, over 80,000 of which due to opioids. One of the more vulnerable populations is US military veterans. Nearly 250,000 military veterans suffer from substance-related disorders (SRD). For those seeking treatment, buprenorphine is prescribed to help treat opioid use disorder (OUD). Urinalysis is currently used to monitor buprenorphine adherence as well as to detect illicit drug use during treatment. Sometimes sample tampering occurs if patients seek to generate a false positive buprenorphine urine test or mask illicit drugs, both of which can compromise treatment. To address this problem, we have been developing a point-of-care (POC) analyzer that can rapidly measure both medications used for treatment and illicit drugs in patient saliva, ideally in the physi-cian’s office. The two-step analyzer employs (1) supported liquid extraction (SLE) to isolate the drugs from the saliva and (2) surface-enhanced Raman spectroscopy (SERS) to detect the drugs. A prototype SLE-SERS-POC analyzer was used to quantify buprenorphine at ng/mL concentrations and identify illicit drugs in less than 1 mL of saliva collected from 20 SRD veterans in less than 20 min. It correctly detected buprenorphine in 19 of 20 samples (18 true positives, 1 true negative and 1 false negative). It also identified 10 other drugs in patient samples: acetaminophen, amphetamine, cannabidiol, cocaethylene, codeine, ibuprofen, methamphetamine, methadone, nicotine, and norbuprenorphine. The prototype analyzer shows evidence of accuracy in measuring treatment medications and relapse to drug use. Further study and development of the system is warranted.

## 1. Introduction

According to the Center for Disease Control, the number of US drug overdose deaths increased from 93,655 in 2020 to 107,622 in 2021 [1,2]. Opioids represented the greatest number of deaths, increasing from an estimated 70,029 in 2020 to 80,816 in 2021. While this crisis affects all classes of people, US military veterans are one of the largest. During Operation Iraqi Freedom and Operation Enduring Freedom, there was a significant increase in the use of opioids prescribed to nearly 700,000 veterans to relieve pain [3]. While the Department of Veterans Affairs (VA) has made efforts to reduce this number [4,5], as of 2020 there remain ~250,000 veterans taking opioids [6], of which 1 in 10 suffer from substance-related disorders (SRDs) [7,8,9].

Buprenorphine is an efficacious and effective medication treatment for opioid use disorder (OUD) with robust evidence supporting its use [10]. It has a very high affinity at the mu opioid receptor, displacing other opioids, and low intrinsic activity. It has 25–40 times the pain-relieving potency of morphine [11] but is considerably less addictive. It was approved by the Federal Drug Administration in 2002 for office treatment by physicians, and is administered as 2, 4, or 8 mg tablets or films; the tablets have slightly slower dissolution times of 5 to 12 min compared to the films [12,13]. Despite the effectiveness of the medication, sometimes diversion and nonadherence occur [14]. In one study, 61% of patients illicitly using buprenorphine indicated they obtained it from people with prescriptions [15]. A number of factors contribute to buprenorphine diversion including buprenorphine’s opioid effects and also the low typical street cost, which is far less than other opioids, such as heroin [16]. Consequently, most clinical protocols (including the VA Clinical Practice Guideline) call for initial and frequent urine drug testing to identify patient discontinuation of medications or any recurrence of illicit drug use. Patients sometimes try to tamper urine drug screens by either simply taking some, but not all of their buprenorphine, or by spiking the urine sample with part of a tablet or film [17,18].

Currently, there are three analysis types for monitoring patient compliance using urine samples; qualitative immunoassay kits (the most common point-of-care tests), semiquantitative immunoassay analyzers, and quantitative liquid or gas chromatography coupled to mass spectrometers (GC- or LC-MS/MS). Aside from using dipsticks or columns built into the sample cups, in the case of immunoassay kits, physicians employ a multidrug assay for drugs, such as cannabinoids, cocaine, opioids, etc. These kits indicate if buprenorphine is above or below a predefined threshold, typically 10 ng/mL urine. However, assays that employ antibodies suffer from false positive rates as high as 25% [19]. The semiquantitative assays employ five or more drug concentration standards, including low and high concentration controls, as well as reagents used to prepare samples, all of which must be refrigerated. Standard and sample preparations and measurements typically take 1 to 2 h [20] and are most often performed in batches using expensive equipment, such as a chemiluminescence analyzer, that require skilled operators in a laboratory. The semiquantitative buprenorphine assays employ DNA sequences instead of antibodies to reduce false positives. Nevertheless, codeine and its metabolites still produce false positives [21]. In contrast to the assays, GC- and LC-MS/MS can be used to measure virtually all drugs and are highly accurate and quantitative, but measurements take several hours, and like semiquantitative measurements, they also require extensive sample preparation, instrument calibration, and skilled operators [19,22,23,24,25].

Consequently, there remains a critical need for a point-of-care (POC) device so that health care personnel can assess SRD patient compliance in outpatient settings. Toward developing such a device, we have been investigating the potential of supported liquid extraction (SLE) to separate the drugs from saliva and surface-enhanced Raman spectroscopy (SERS) to both identify and quantify the drugs [26,27,28,29]. The expected success of this approach is based on the extreme sensitivity of SERS [30,31], the ability to measure very small samples, such as 1 mL of saliva, and the ability to identify molecular structures of drugs through the rich vibrational information provided by Raman spectroscopy [32]. Furthermore, saliva represents an ideal sample medium, since collection is noninvasive and can be performed in the presence of health care personnel, eliminating the chance of sample tampering. Most importantly, it has been reported that sublingually administered buprenorphine concentrations in saliva can average ~50 ng/mL for an 8 mg/day dose [33], similar to urine at an average of 160 ng/mL for an average 16 mg/day dose [17].

Previously, we established the ability of a laboratory SLE-SERS analyzer to perform measurements of buprenorphine in patient saliva samples. However, the sensitivity was limited to 1 µg/mL, and multidrug analysis was limited by a significant background produced by the SERS substrate [34]. Here, we present the development of a portable SLE-SERS-POC prototype analyzer and its use to detect illicit drugs and quantify buprenorphine at ng/mL concentrations extracted from <1 mL of saliva samples provided by 20 SRD veterans undergoing treatment. The primary objective of this study was to quantify buprenorphine in patient saliva collected in a physician’s office as a potentially better measure of adherence. The new design improved sensitivity by a factor of ~50. A secondary objective was to identify other drugs in such samples. Taken together, this information could improve diagnosis and patient treatment. However, the second objective was best performed first to determine if there were any spectral interferences from other drugs in the samples that would hinder quantitation of buprenorphine.

## 2. Results

### 2.1. Drug Identification

Twenty patient saliva samples were prepared and measured by the SLE-SERS-POC prototype analyzer as described in the Methods section. The drug contributions to each sample spectrum were determined in two steps. First, all the library spectra were ranked in terms of the closest match to the sample spectrum. Second, the sample spectrum was fit with weighted contributions of the closest matched spectra. In both cases, the analyses were confined to the 400 to 1800 cm^−1^ Raman spectral range, such that the analyses could be performed automatically. The Correlation algorithm [35] was used to match the sample spectrum to library spectra in terms of a hit quality index (HQI), where an exact match equals 0, and a complete mismatch equals 1 [36]. The first derivatives of the sample and library spectra were used to remove spectral background effects prior to matching. For Sample 1, the top three HQIs scores were 0.335 for nicotine, 0.645 for buprenorphine, and 0.945 for methadone. The latter drug score is considered a near perfect mismatch and was excluded from the analysis (Figure 1A,B and Appendix A). Nicotine, characterized by narrow peak at 1030 cm^−1^, due to a phenyl ring vibration, was detected in five samples. This was not surprising as several patients indicated they were smokers on their demographics form. Each sample spectrum was then fit using the relative spectral contributions of each drug that had an HQI score less than 0.7, setting the total to 100%. This reduced the contributions to five drugs or less. Only those drugs, whose spectra contributed 5% or more, were included in the analysis, and are reported as whole numbers. The spectral fit software indicated that the Sample 1 spectrum was composed of 54% buprenorphine and 46% nicotine (Figure 1C). While most of the samples could be fit following this procedure, some were more challenging. For example, the top HQIs for the first derivative spectra of Sample 13 were 0.096 for ibuprofen and 0.366 for norbuprenorphine, all other library drugs were greater than 0.7. A satisfactory fit for the Sample 13 spectrum was obtained using 95% ibuprofen and 5% norbuprenorphine (Figure 1D). Nevertheless, the spectral fit did not capture all of the Sample 13 spectra peaks, indicating that other chemicals, biologicals, or drugs were present, but were not in the spectral library and therefore were not included in the analysis.

The remaining 18 sample spectra, fit using the same procedure, are shown in Figure 2. The drug contributions, totaling 100% for each sample, and the corresponding urinalysis results for all 20 samples, are listed in Table 1. Buprenorphine was detected in 18 patient samples, 5 at 100%, 4 additional samples greater than 70%, 7 additional samples greater than 50%, and 2 samples less than 50%. Other drug contributions detected by SERS were as follows (Figure 3): nicotine in 5 samples, norbuprenorphine in 4 samples, acetaminophen in 4 samples, cannabinoids in 4 samples (as cannabidiol), cocaethylene in 2 samples, methadone in 2 samples, and amphetamine, codeine, ibuprofen, and methamphetamine in 1 sample each. It was also noted that Samples 7, 13, 18, and 20 had contributions from unknown chemicals, biologicals, or drugs not in the spectral library. It is worth noting that naloxone was not detected in any of the samples, undoubtedly due to its poor absorption from sublingual tablets [37].

Compared to urinalysis, SERS correctly identified buprenorphine in 19 of 20 samples; 18 true positives, 1 true negative (Sample 16), and 1 false negative (Sample 13), which contained norbuprenorphine. The following additional comparisons of SERS to urinalysis were noted (Table 1). Cannabinoids were detected by SERS in only four of the six positive urinalysis samples. Amphetamine was detected by SERS in Sample 3 and by urinalysis. Methamphetamine was detected in Sample 2 by SERS, but not by urinalysis. Methadone was detected in Samples 10 and 14 by SERS, but not by urinalysis. Yet methadone was detected by urinalysis in Sample 12, which was not detected by SERS. Cocaine was detected in Samples 17 and 20 by urinalysis, while it was detected as cocaethylene in Samples 7 and 20 by SERS. These discrepancies are likely due to the fact that urinalysis detects the benzoylecgonine metabolite, while SERS detects cocaethylene metabolite. In addition, their relative concentrations may be significantly different in saliva and urine.

Opioids were detected by SERS in three samples, but none matched the three samples that tested positive for opioids by urinalysis. This discrepancy and others indicated in Table 1 may be due to the differences in saliva versus urine metabolites, as well as relative concentrations.

### 2.2. Buprenorphine Quantitation

The second objective required identifying the best buprenorphine spectral peak to use for quantitation. Two factors were examined: (1) the pH spectral dependence of buprenorphine and (2) the spectral interferences by the other drugs in the samples. The SERS intensity of drugs often has a pH dependence, due to the fact that protonated and deprotonated molecules interact with the surface plasmon of gold nanoparticles to varying degrees [38]. This dependence also influences the molecular-to-gold surface orientation and hence the relative intensities of the various functional group spectral peaks. Consequently, the pH dependence of buprenorphine samples at 100 µg/mL, adjusted from pH 5 to 8 by adding HCl or NaOH, was measured by SERS. The following SERS peaks were observed and have been assigned as follows: 505 cm^−1^ to a-/c-ring CH bending, 638 cm^−1^ to a-ring C=CH out-of-plane bending, 735 cm^−1^ to out-of-plane C=O bending, 835 cm^−1^ to c-ring CH bending, 1210 cm^−1^ to c-ring CCC out-of-plane bending, 1310 cm^−1^ to d-ring piperidine CH stretch, and 1590 cm^−1^ to c-ring CC stretching (Figure 4 and Figure 5).

It was found that the 835 cm^−1^ peak was the most intense at pH 7 and the 1210 cm^−1^ peak was the most intense at pH 6, while the 638 cm^−1^ peak, although modest in intensity, was relatively insensitive from pH 5 to 8 (Figure 4A). An examination of the 36 drugs in Table 2 revealed only five drugs that contained peaks in the region of the 638 cm^−1^ buprenorphine peak; heroin, codeine, diazepam, norbuprenorphine, and hydrocodone (Figure 4B). These drugs were included in the analysis of the 20 samples described above. Of these drugs, only norbuprenorphine and codeine were detected in the 20 samples. Norbuprenorphine was detected in Samples 5, 6, 8, and 13. There is no confusion that both buprenorphine and norbuprenorphine are present, because the analysis employs the full spectrum (400 to 1800 cm^−1^) and each drug has unique peaks that were detected: 835 and 1445 cm^−1^ for buprenorphine and 1015, 1135, and 1635 cm^−1^ for norbuprenorphine. Codeine was detected in Sample 9 and also has spectral peaks that differentiate it from buprenorphine at 535 and 1255 cm^−1^. Consequently, the 638 cm^−1^ peak was used to quantify buprenorphine in the samples.

Next, buprenorphine samples were prepared in purchased, deidentified pooled saliva at 6.25, 12.5, 25, 50, 100, and 200 ng/mL, extracted and reconstituted as described in Method 3, and measured by SERS to produce a calibration curve. The baseline of the spectra was set to 0 at 665 cm^−1^, and the 638 cm^−1^ peak height was plotted as a function of the prepared concentrations and fit with a straight line: [BUP] = 0.029 × Peak Height + 0.19, with an R^2^ value of 0.99 (Figure 5C and Appendix A).

The equation was then used to calculate the concentration for each of the 20 samples using the SERS measured and baseline-corrected: 638 cm^−1^ peak height (Table 1, right-most column, and Appendix A). The concentrations for samples containing norbuprenorphine and codeine, as described above, were corrected using the spectral fit percent results for these samples. For example, the Sample 5 buprenorphine concentration was reduced from 39 to 35 ng/mL, since the 638 cm^−1^ peak was composed of 89% buprenorphine. Similarly, for Sample 9, containing codeine, the buprenorphine concentration was reduced from 80 to 45 ng/mL (56%). The uncorrected concentrations are shown in parentheses in Table 1.

### 2.3. Analytical Figures of Merit

The analysis reproducibility, limit of detection (LOD), and limit of quantitation (LOQ) were also determined for the measurement procedure. Analysis reproducibility, encompassing sample preparation, extraction, reconstitution, and measurement, was determined by measuring nine independently prepared 50 ng/mL buprenorphine samples, consisting of 40 µL drops, ~5 mm in diameter, deposited on glass slides. It was found that the percent standard deviation for the 638 cm^−1^ peak height was 3.5% (Figure 6 and Appendix A, Table 3). Note that SERS measurement repeatability, performed by measuring nine positions of a single 50 ng/mL, 40 µL drop, yielded a percent standard deviation of ~1%, indicating that the colloid and buprenorphine were evenly distributed in the sample.

Lastly, the LOD and LOQ, were calculated as 1.4 and 4.6 ng/mL based on signal-to-noise ratios (S/N) of 3 and 10, respectively, according to: LOD = sample concentration/[(S/N)/3] and LOQ = sample concentration/[(S/N)/10]. The 50 ng/mL sample was used for the calculation. The signal was taken as the 638 cm^−1^ peak height, baseline-corrected from 610 to 665 cm^−1^ (S = 1705), and the root mean squared (rms) noise (15.6) between 1810 and 1865 cm^−1^ was used (Figure 7). The latter spectral region was chosen as it contains only background noise, while the width was selected to match the peak width.

## 3. Discussion

The main goal of this study was to test the ability of an SLE-SERS-POC prototype analyzer to determine patient compliant use of buprenorphine. In this regard, the prototype performance was very good, correctly identifying 18 adherent patients and 1 not adherent patient to medication in agreement with urinalysis. The prototype only misidentified one patient sample as nonadherent who was adherent according to the urinalysis results. It is possible that the patient spiked their urine sample with buprenorphine. While the prototype provided quantitative buprenorphine concentration for all of the samples, there was no relation between the patient SERS-based saliva concentrations and their administered dosage. This could be caused by two factors: variability of the patient’s metabolism, or more likely, the time the saliva samples were collected with respect to when the patient took their dose. It has been shown that the saliva concentration is as high as 1000 ng/mL within the first hour after sublingual administration of a 1 mg tablet and does not reach a steady state until 10 h after administration due to “holding” in the oral cavity and buccal permeability [33,39] In fact, some patients supplied saliva samples within an hour after taking a dose. Furthermore, the saliva concentration is relatively stable from 10 to 24 h, suggesting that the best time to perform a saliva measurement would be right before the tablet is administered. In addition, measurement of a patient at the same time for several days would be advantageous to setting dosage based on metabolism and thereby improve patient performance. It is also worth noting that the prototype LOD, LOQ, and R^2^ values are similar to LC-MS/MS measurements of buprenorphine in saliva at 5 ng/mL, 10 ng/mL, and 0.9986, respectively [23].

## 4. Materials and Methods

### 4.1. Materials 1: Purchased Materials

All chemicals and solvents used to prepare samples, colloids, and perform extractions were obtained from Sigma–Aldrich (St Louis, MO, USA). The drugs used to prepare the spectral library were purchased as 1 mg/mL methanol forensic samples from the same supplier (Table 2). A quantity of 330 cu. ft. of ultrahigh purity nitrogen pressurized gas was provided from a tank (Airgas, Hartford, CT, USA). Deidentified pooled saliva was purchased from Lee Biosolutions (Maryland Heights, MO, USA). Saliva collector/dropper tubes were purchased from Medimpex United (Bensalem, PA, USA). Supported liquid extraction (SLE) columns, containing 87.7% particles between 106 to 180 µm, with a 400 µL volume capacity, obtained from Biotage (model ISOLUTE SLE+ 400, Salem, NH, USA), were used in conjunction with a Resprep QR-12 column vacuum manifold (Restek, Bellefonte, PA, USA) and a 0.2 HP Air Cadet vacuum pump (model 420-3901, ThermoFisher, Boston, MA, USA). Glass support slides were obtained from VWR (Radnor, PA, USA). The Raman spectrometer employed a 785 nm laser and a −15 °C cooled, 2048-pixel Si detector (model WP-785-A-SR-L-50, Wasatch Photonics, Morrisville, NC, USA). A laptop was used to collect and analyze the Raman spectra (Inspiron, Dell, Round Rock, TX, USA). The components of the prototype SLE-SERS-POC analyzer are shown in Figure 8.

### 4.2. Materials 2: Prepared Materials

The gold colloid solution used for SERS measurements was synthesized following a modified Lee–Meisel method [40]. Briefly, 240 mg of gold chloride (HAuCl_4_•3H_2_O) was dissolved in 500 mL of water and heated to 100 °C, at which temperature 50 mL of 1% sodium citrate was added and then allowed to boil for 1 hr. The gold colloids have a shelf life of over a month and were prepared in advance. The forensic drug samples were diluted to 100 µg/mL using distilled water. Twenty microliter aliquots of these diluted drug samples were mixed with 20 µL of the gold colloids. Similarly, a concentration series of buprenorphine was prepared by diluting a 1 mg/mL methanol forensic sample to 5 ng/mL in distilled water. For each concentration, 200 µL of buprenorphine in water was added to 200 µL of deidentified pooled saliva.

### 4.3. Materials 3: Patient Samples

VA Connecticut Healthcare System (VACHS, West Haven, CT, USA) patients being treated for OUD using buprenorphine who were already providing urine samples for analysis as part of a larger study were recruited to provide saliva samples in accordance with IRB Protocol 00008942 (Chesapeake IRB, Inc., Columbia, MD, USA). Twenty volunteer patients went through an informed consent process, stated that they understood this study, and signed the consent and Health Insurance Portability and Accountability Act documents. Most of the patients were taking buprenorphine for at least 2 weeks prior to providing a saliva sample, which was collected within 2 h of the urine samples. The patients also provided information regarding drug use for the previous 2 weeks. Buprenorphine was administered sublingually once or twice a day as Suboxone containing 2, 4 or 8 mg of buprenorphine and 0.5, 1 or 2 mg naloxone, respectively. Total daily patient doses were 8, 12, 16, 20 or 24 mg buprenorphine. Ten minutes prior to saliva sample collection, the patients were instructed to rinse their mouth out with bottled water. Sample collection was performed by spitting into plastic tubes until 1 to 2 mL of saliva was obtained. Everyone on buprenorphine had their dose the day the sample was collected and took a dose within a few hours before the saliva sample was collected. The saliva samples were sealed and frozen until saliva analysis was performed at Real-Time Analyzers (RTA).

The general demographics for the 20 patients were as follows (Table 4): all male, 12 Caucasian, 5 African American, 2 Hispanic, and 1 declined; 2 under age 40, 6 between 40 and 49, 4 between 50 and 59, and 7 between 60 and 65 years of age. The 20 samples were all collected over the course of one week.

### 4.4. Method 1: Urine Toxicology

Urine samples provided by patients were delivered to an on-site VACHS clinical laboratory. The samples were analyzed by a standard multiplexed sample and reagent handling system coupled to an electrogenerated chemiluminescence analyzer (e.g., Roche Hitachi 6001). Immunoassays consisted of standard ruthenium functionalized drug-specific antibodies on magnetic beads such that separation could be accomplished using an electrode to capture the beads and generate chemiluminescence. Magnetic beads, functionalized with a DNA sequence, were used to bind buprenorphine instead of an antibody, as used for all other drugs. The intensity of the luminescence signal was compared to a positive cut-off calibrant sample selected for each of the 9 drugs (Table 5).

### 4.5. Method 2: Liquid Extraction

A 200 µL saliva sample mixed with 200 µL of distilled water was added to a SLE column attached to the vacuum pump. The sample was adsorbed onto the support by applying a negative pressure of 15 inch of Hg for 1 sec. After a 5 min wait, 2 sequential aliquots of 900 µL dichloromethane were drawn through the support, first using gravity for 5 min, then again using −15 inch of Hg for 1 min. The collected sample was dried under a gentle stream of nitrogen for 5 min, reconstituted using 40 µL of the gold colloid solution, of which a 10 µL drop was deposited a onto a glass slide, which in, turn was placed in the sample compartment of the Raman spectrometer for analysis.

### 4.6. Method 3: Raman Spectroscopy

The 5 µL aliquots of library drugs, buprenorphine concentration series, and patient saliva samples mixed with 5 µL of the gold colloids were each deposited on glass slides and placed into the sample compartment above the focal point of the 785 nm laser of the Raman spectrometer (Wasatch Photonics, model FPR-785-WS, Orlando, FL, USA). Spectra were recorded from ~200 cm^−1^ to 2300 cm^−1^ with ~20 cm^−1^ resolution (peak width at half height), and peak positions are reported to the nearest 5 cm^−1^ (except the 638 cm^−1^ buprenorphine peak). A laptop computer was used to control the laser power, acquisition time, and perform spectral analysis. Each spectrum consisted of a 1 sec acquisition using ~40 mW of 785 nm laser excitation focused to ~200 µm at the sample. RTA’s Chem-ID and S-Quant software were used to identify drugs and quantify buprenorphine in the samples, respectively. A complete surface-enhanced Raman spectral analysis of the drugs described here has been published [32].

## 5. Conclusions

The SLE-SERS-POC prototype analyzer successfully quantified buprenorphine in samples in less than 20 min for 20 VA patients. A simple supported liquid extraction method was successfully developed to isolate the drugs from the saliva samples for analysis by surface-enhanced Raman spectroscopy. Semiautomated spectral analysis, employing a spectral library, identified 25 of 30 drugs in the samples. This included the identification of six drugs without prior knowledge of their presence in the 20 saliva samples. Furthermore, the presence of these additional drugs did not interfere with the measurements. Nevertheless, the method could be improved by using a library that includes only those drugs that could be reasonably expected in patient samples. We believe that an SLE-SERS-POC production analyzer could greatly improve patient adherence by eliminating drug spiking and aid physicians in setting dosage and monitoring buprenorphine and thereby improve treatment. Future work will focus on determining the best method to collect saliva, (e.g., passive drool or swab) and automating the sample extraction and analysis software. A next-stage prototype will be used to perform initial clinical trials that include comparing SLE-SERS-POC analysis in saliva to urine samples collected from patients and measured by LC-MS/MS.

## Figures and Tables

**Figure 1 molecules-28-02010-f001:**
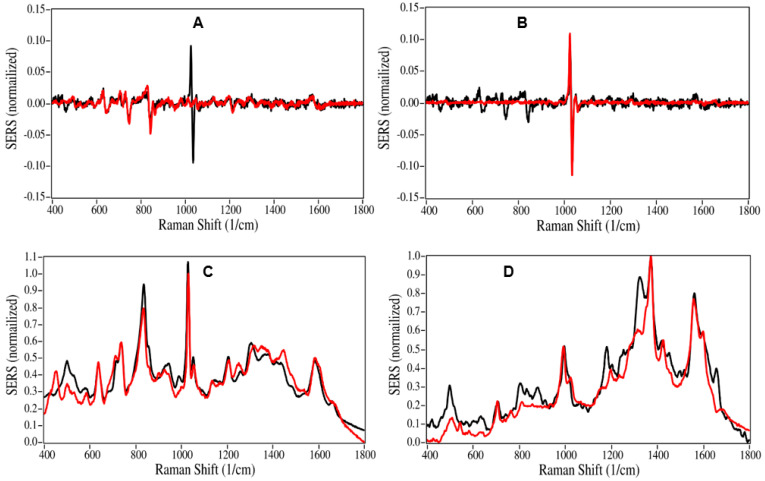
SERS of Sample 1 first derivative (black) overlaid with (**A**) first derivative of buprenorphine (red), (**B**) overlaid with first derivative of nicotine (red). (**C**) Overlay of Sample 1 spectrum (red) on the predicted spectrum composed of 54% buprenorphine and 46% nicotine (black). (**D**) Overlay of Sample 13 spectrum (red) on predicted spectrum composed of 95% ibuprofen and 5% norbuprenorphine (black). Conditions: 100 mg/L sample in water, 40 mW at 785 nm, 1 sec acquisition.

**Figure 2 molecules-28-02010-f002:**
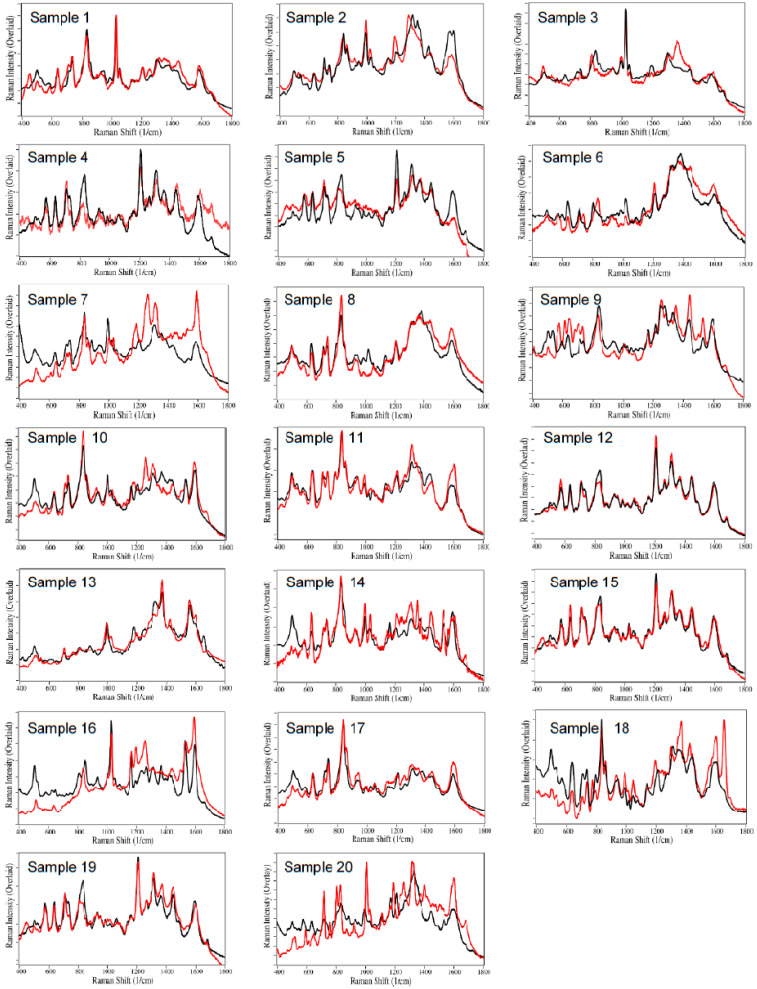
SERS of VACHS Samples 1 to 20. Actual spectra (red) are fit by the S-Quant software (black). Spectral conditions as in Figure 1.

**Figure 3 molecules-28-02010-f003:**
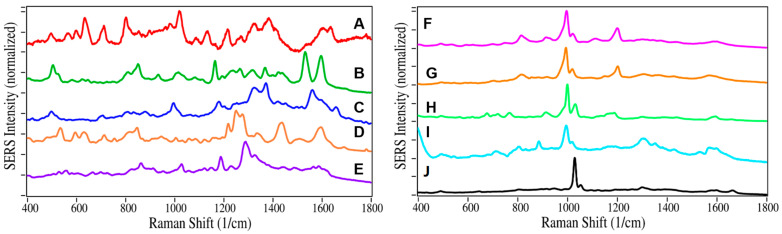
SERS of drugs other than buprenorphine identified in the various samples: (**A**) norbuprenorphine, (**B**) acetaminophen, (**C**) ibuprofen, (**D**) codeine, (**E**) cannabidiol, (**F**) amphetamine, (**G**) methamphetamine, (**H**) methadone, (**I**) cocaethylene, and (**J**) nicotine. All samples measured at 100 µg/mL in water. Conditions as in Figure 1.

**Figure 4 molecules-28-02010-f004:**
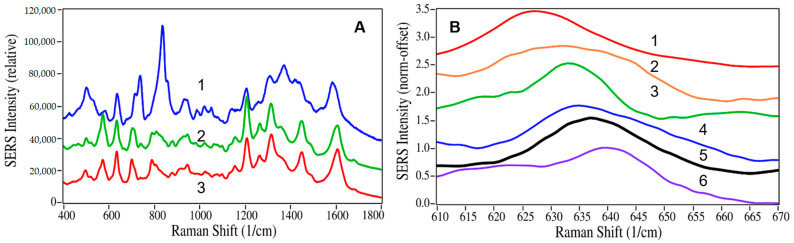
(**A**) Spectra of 100 µg/mL buprenorphine in water adjusted to (1) pH 7, (2) pH 6, and (3) pH 5. (**B**) Spectra of (1) heroine, (2) codeine, (3) diazepam, (4) norbuprenorphine, (5) buprenorphine, and (6) hydrocodone in the region of the buprenorphine 638 cm^−1^ peak. Conditions as in Figure 1.

**Figure 5 molecules-28-02010-f005:**
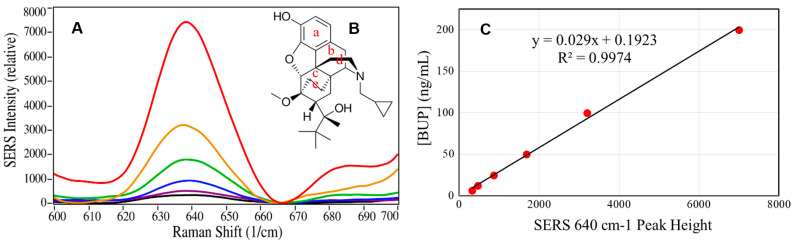
(**A**) Spectra of buprenorphine 638 cm^−1^ peak, baseline-corrected at 665 cm^−1^, in pooled saliva, extracted and reconstituted at 6.25, 12.5, 25, 50, 100, and 200 ng/mL, (**B**) buprenorphine molecular structure, and (**C**) plot of buprenorphine concentration as a function of baseline-corrected 638 cm^−1^ peak heights. Conditions as in Figure 1.

**Figure 6 molecules-28-02010-f006:**
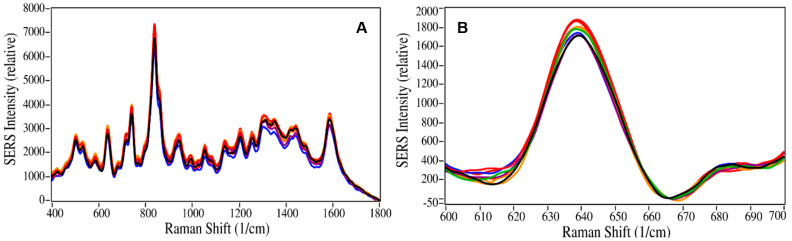
Nine overlaid SERS of 50 ng/mL buprenorphine extracted from prepared pooled saliva samples, (**A**) baseline-corrected at 1800 cm^−1^, and (**B**) baseline-corrected at 665 cm^−1^. Conditions as in Figure 1.

**Figure 7 molecules-28-02010-f007:**
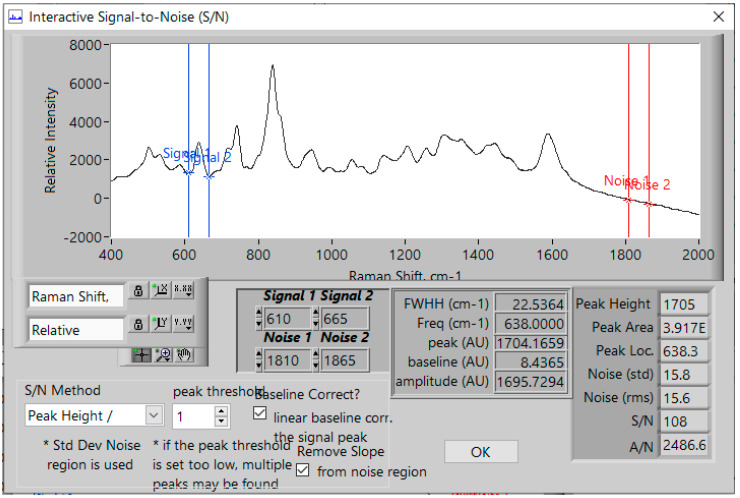
Image of Real-Time Analyzers’ S/N program used to calculate the S/N of 109 for 50 ng/mL and an LOD of 1.4 ng/mL and an LOQ of 4.6 ng/mL buprenorphine using the 638 cm^−1^ peak and the rms noise between 1810 and 1865 cm^−1^. Conditions as in Figure 1.

**Figure 8 molecules-28-02010-f008:**
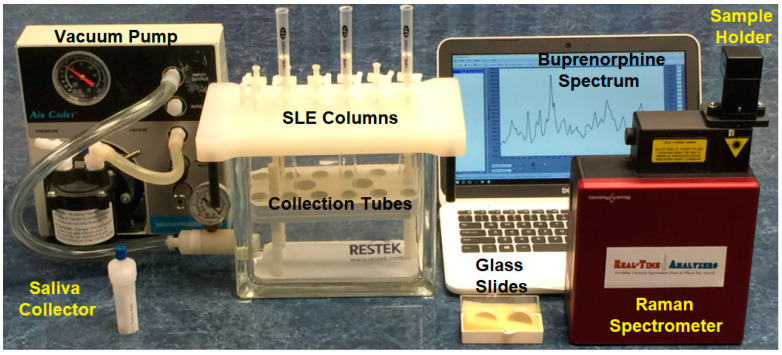
Photograph of the SLE-SERS-POC prototype analyzer.

**Table 1 molecules-28-02010-t001:** Sample number, daily dose, urinalysis, SERS analysis (drugs identification and percentages), and SERS quantitation of buprenorphine in saliva samples.

Sample and mg/day		Urinalysis					SERS Analysis (ID and %)							Saliva-SERS ng/mL
		BUP^a^	CAN	COC	OPI	Other	BUP	CAN	OPI/Other	NOR	NIC	ACE	UNK	
1	24	Y					53				47			111
2	20	Y	Y				53	24^b^	MAMP 23					164
3	8	Y	Y			AMP	23	24^b^	AMP 7		46			1
4	24	Y					100							34
5	20	Y					89			11				35 (39)
6	16	Y					68			32				18 (26)
7	20	Y					61		COC-ET 39				UNK	75
8	20	Y					92			8				48 (52)
9	16	Y	Y				56	10^c^	COD 15			19		45 (80)
10	8	Y	Y				65		MDON 21			14		121
11	24	Y					100							88
12	8	Y			Y	MDON	100							55
13	24	Y	Y				0		IBU 95	5			UNK	0 (6)
14	24	Y					61		MDON 23		11	5		147
15	8	Y					85				15			206
16	8	NO			Y		0				66	34		0
17	20	Y		Y			100							55
18	8	Y			Y		>70						UNK	38
19	24	Y					100							98
20	12	Y	Y	Y			38	56^c^	COC-ET 6				UNK	20

The uncorrected buprenorphine concentrations are shown in parentheses. a = Abbreviations: ACE-acetaminophen, AMP-amphetamine, BUP-buprenorphine, CAN-cannabinoids, COC-cocaine, COC-ET-cocaethylene, COD-codeine, MAMP-methamphetamine, MDON-methadone, MOR-morphine, NIC-nicotine, NOR-norbuprenorphine, OPI-opioids, UNK-unknown biochemical, chemical, or drug, Y = yes, b = cannabinol, c = delta9 THC.

**Table 2 molecules-28-02010-t002:** List of 36 drugs included in the SERS library. All drugs were measured at 100 µg/mL.

acamprosate	bupropion	codeine	ibuprofen	methadone	norbuprenorphine
acetaminophen	caffeine	Δ-THC	LSD	methamphetamine	nordiazepam
amphetamine	cannabidiol	diazepam	MDA	methylphenidate	oxazepam
aspirin	cannabinol	fentanyl	MDMA	morphine	oxycodone
benzoylecgonine	cocaethylene	heroin	meperidine	naloxone	PCP
buprenorphine	cocaine	hydrocodone	mescaline	nicotine	secobarbital

**Table 3 molecules-28-02010-t003:** Corresponding buprenorphine 638 cm^−1^ peak heights.

Repeat SLE-SERS		
Sample	638 cm^−1^ Ht	Calc. Conc.
1	1754	51.1
2	1755	51.1
3	1682	49.0
4	1609	46.9
5	1634	47.6
6	1649	48.0
7	1727	50.3
8	1709	49.8
9	1606	46.8
AVE	1680.6	48.9
STD DEV	58.9	1.7
% STD DEV	3.5	3.5

**Table 4 molecules-28-02010-t004:** Demographic information for 20 enrolled patients (see Appendix A for additional details).

Patient Demographic Information								
	African American	American Indian and Alaskan	Asian	Caucasian	Hispanic	Native Hawaiian and Pacific Islander	Other	Total
Male	5	0	0	12	2	0	1	20
Female	0	0	0	0	0	0	0	0
Total	5	0	0	12	2	0	1	20

**Table 5 molecules-28-02010-t005:** Drugs measured by chemiluminescence and their calibrants and positive cut-off values.

Drug	Calibrant	Positive Detection Cut-Off Value
Buprenorphine	Buprenorphine	10 ng/mL
Amphetamines	d-methamphetamine	1000 ng/mL
Barbiturates	Secobarbital	200 ng/mL
Benzodiazepines	Oxazepam	200 ng/mL
Cannabinoids	Delta-9-THC	50 ng/mL
Cocaine	Benzoylecgonine	300 ng/mL
Opiates	Morphine	300 ng/mL
Methadone	Methadone	300 ng/mL
Oxycodone	Oxycodone	100 ng/mL

## Data Availability

Not applicable.

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
