# Peer review of "Analysis of Drugs in Saliva of US Military Veterans Treated for Substance Use Disorders Using Supported Liquid Extraction and Surface-Enhanced Raman Spectral Analysis"

_molecules, 2023, doi:10.3390/molecules28052010_

Round 1
Reviewer 1 Report
Manuscript reports on development of combined Supported liquid extraction (SLE) and Surface-enhanced Raman spectroscopy (SERS) approach for quantitative analysis of buprenorphine and other drugs. The problem is important and presented data suggest that such approach is perspective. In the manuscript only the first step for solving this problem related with development of the prototype analyzer is presented and possible discrepancies are discussed. However, in my opinion, this work in not suitable for “Molecules” because no molecular analysis is presented; other journals, such as “BioMed”, “Biomedicines”, “Journal of Clinical medicine” or “Diagnostics” are more suitable.
Minor points:
1) More detailed description of Raman spectrometer is required including the manufacturer details and diameter of laser spot on the sample.
2) The molecular structure of buprenorphine drug should be provided in the main manuscript or in the Supporting information file.
Author Response
We added:
Raman spectrometer (Wasatch Photonics, model FPR-785-WS, Orlando, Florida).
Each spectrum consisted of a 1 sec acquisition using ~40 mW of 785 nm laser excitation focused to ~200 µm at the sample.
I labelled the primary peaks in Figure 5A and added the structure in Figure 6A, I also added peak assignments in the text.
As to the "fit" to the Molecules journal, they asked if I had anything that might fit. I provided them an outline. They said please submit, so I did.
Reviewer 2 Report
This study demonstrates a prototype analyzer combining supported liquid extraction (SLE) and surface enhanced Raman spectroscopy (SERS) to treat, then measure saliva samples, that was carried out in a compact diagnostic system as a point-of-care test. The semi-automated spectral analysis made the identification of 11 different drugs possible that were included in a spectral library. The authors have shown a quantitation procedure for buprenorphine from SER spectra. The illustrated results are convincing and promising. The analyzer can be faster than the commonly used urinalysis.
The manuscript is of high quality, considering the novelty, methodology, and results.
Grammatical failures
Line 124 were added → was added
Line 129 were added → was added
Line 274 have a pH dependence → has a pH dependence
Line 279 were measured → was measured
Line 387 used to determine → was used to determine
Author Response
Thank you pointing out these errors, we fixed them.
Reviewer 3 Report
In this manuscript, the authors reported a novel SLE-SERS-POC method which can rapidly measure both medications used for treatment and illicit drugs in patient saliva. The results are interesting and the manuscript is well organized. There some points should be addressed before acceptance for publication.
1. There are some typos in the manuscript. For example, “A complete spectral analysis of the library SER spectra has been published.” In Page 5. Is it “SERS” rather than SER.
2. In the supplementary information, Figures S1c and d should be replaced by a high-resolution images. They are not clear in this stage.
3. The description of “The LOD and LOQ were 1.4 (50/(108/3)) and 4.2 ng/mL, respectively.” In Page 10. The calculation method and equation for LOD and LOQ should be described in details and clearly. The detailed description is recommended to provide in the supplementary information file.
4. The Raman peaks, e.g. 638 cm-1, should be well marked and identified for better understand by the audience.
Author Response
- There are some typos in the manuscript. For example, “A complete spectral analysis of the library SER spectra has been published.” In Page 5. Is it “SERS” rather than SER.
We changed the sentence as follows:
“A complete surface-enhanced Raman spectral analysis of the drugs described here has been published.” (ref 32)
- In the supplementary information, Figures S1c and d should be replaced by a high-resolution images. They are not clear in this stage.
All of the supplementary spectral figures are screen captures from analysis software. I did enlarge the figures. I hope this helps.
- The description of “The LOD and LOQ were 1.4 (50/(108/3)) and 4.2 ng/mL, respectively.” In Page 10. The calculation method and equation for LOD and LOQ should be described in details and clearly. The detailed description is recommended to provide in the supplementary information file.
We rewrote the LOD, LOQ paragraph as follows:
Lastly, the LOD and LOQ, were calculated as 1.4 and 4.6 ng/mL based on signal-to-noise ratios (S/N) of 3 and 10, respectively, according to: LOD = sample concentration/[(S/N)/3] and LOQ = sample concentration/[(S/N)/10]. The 50 ng/mL sample was used for the calculation. The signal was taken as the 638 cm-1 peak height, baseline corrected from 610 to 665 cm-1 (S =1705), and the root mean squared (rms) noise (15.6) between 1810 and 1865 cm-1was used (Figure 8). The latter spectral region was chosen as it contains only background noise, while the width was selected to match the peak width.
The Raman peaks, e.g. 638 cm-1, should be well marked and identified for better understand by the audience.
We labelled the peaks in Figure 5A, and added a structure in Figure 6A, as well as and vibrational assignments in the body of the text (ref 32).
Reviewer 4 Report
The authors present the method for determination of buprenorphine in saliva using Raman spectroscopy after supported liquid extraction (SLE). The prototype equipment was evolved and the Raman spectra were obtained via surface-enhanced Raman spectroscopy (SERS) with gold colloid solution. The study seems to be a continuation and further development of the previous one published in 2017:
“Farquharson S, Dana K, Shende C, Gladding Z, Newcomb J, et al. (2017) Rapid Identification of Buprenorphine in Patient Saliva. J Anal Bioanal Tech 8: 368. doi: 10.4172/2155-9872.1000368”
During current study the authors selected another wavelength as buprenorphine detection point (638 1/cm currently vs 835 1/cm previously). The previous work was not mentioned however, since the authors achieved improved selectivity, sensitivity and the method became quantitative it would be worth to underline the added value of the study.
The selectivity of the method as well as linearity, sensitivity and repeatability was tested. The authors discussed advantages and disadvantages of the method and compared the results with well-established urinalysis. The study is clearly presented and the conclusions are justified.
The following should be added/clarified/corrected:
1. 1. The introduction should mention what was done before and potential improvements, changes and advancements should be clarified.
2. 2. Details about SLE column e.g. brand name, column size and bed capacity should be given.
3. 3. Table 4 is unclear – it is hard to guess which column belongs to each of the three main column headers
4. 4. Figure 4 - spectrum J is wrongly signed as F.
5. 5. At least a word “validation” should be added to the part describing linearity, repeatability, LOQ and LOD verification. I wonder if accuracy/recovery can be also drawn from the tests already performed.
Author Response
- The introduction should mention what was done before and potential improvements, changes and advancements should be clarified.
We added to the following to the introduction. Previously, we established the ability of a laboratory SLE-SERS analyzer to perform measurements of buprenorphine in patient saliva samples. However, the sensitivity was limited to 1 µg/mL, and multi-drug analysis was limited by a significant background produced by the SERS substrate. Here we present the development of a portable SLE-SERS-POC prototype analyzer and its use to detect illicit drugs and quantify buprenorphine at ng/mL concentrations extracted from <1 mL of saliva samples provided by 20 SRD veterans undergoing treatment. The primary objective of this study was to quantify buprenorphine in patient saliva collected in a physician’s office as a potentially better measure of adherence. The new design improved sensitivity by a factor of ~50.
- Details about SLE column e.g. brand name, column size and bed capacity should be given.
We added the following
Supported liquid extraction (SLE) columns, containing 87.7% particles between 106 to 180 µm, with a 400 µL volume capacity, obtained from Biotage (model ISOLUTE SLE+ 400, Salem, NH), were used in conjunction with a Resprep QR-12 column vacuum manifold (Restek, Bellefonte, PA) and a 0.2 HP Air Cadet vacuum pump (model 420-3901, ThermoFisher, Boston, MA).
- Table 4 is unclear – it is hard to guess which column belongs to each of the three main column headers
We added some thicker lines for header breaks in the table.
- Figure 4 - spectrum J is wrongly signed as F.
Fixed.
- At least a word “validation” should be added to the part describing linearity, repeatability, LOQ and LOD verification. I wonder if accuracy/recovery can be also drawn from the tests already performed.
As indicated by the last sentence of the Conclusion, we hope to validate the method in the future.
Round 2
Reviewer 1 Report
The manuscript was improved; all concerns raised by the Reviewer were addressed. The manuscript can be published in current form.
Reviewer 4 Report
no further comments